# Prefrontal projections to the thalamic nucleus reuniens mediate fear extinction

Karthik R. Ramanathan [1], Jingji Jin[1], Thomas F. Giustino [1], Martin R. Payne[1] & Stephen Maren [1]

The thalamic nucleus reuniens (RE) receives dense projections from the medial prefrontal cortex (mPFC), interconnects the mPFC and hippocampus, and may serve a pivotal role in regulating emotional learning and memory. Here we show that the RE and its mPFC afferents are critical for the extinction of Pavlovian fear memories in rats. Pharmacological inactivation of the RE during extinction learning or retrieval increases freezing to an extinguished conditioned stimulus (CS); renewal of fear outside the extinction context was unaffected. Suppression of fear in the extinction context is associated with an increase in c-fos expression and spike firing in RE neurons to the extinguished CS. The role for the RE in suppressing extinguished fear requires the mPFC, insofar as pharmacogenetically silencing mPFC to RE projections impairs the expression of extinction memory. These results reveal that mPFC-RE circuits inhibit the expression of fear, a function that is essential for adaptive emotional regulation.

---

[1] Department of Psychological and Brain Sciences and Institute for Neuroscience, Texas A&M University, College Station, TX 77843, USA. These authors contributed equally: Karthik R. Ramanathan, Jingji Jin. Correspondence and requests for materials should be addressed to S.M. (email: maren@tamu.edu)

 

Learning to contend with threats in the environment is essential to survival. It allows animals, whether rats or humans, to anticipate harm and organize appropriate defensive behaviors in response to threat. However, aversive learning can become maladaptive and lead to pathological conditions such as panic disorder, anxiety, and post-traumatic stress disorder to name a few[1,2]. Of course, fear memories are evolutionarily programmed to be rapidly acquired, temporally enduring, and broadly generalized across both familiar and novel contexts. In contrast, procedures that reduce fear and anxiety, such as exposure therapy, tend to produce fear suppression that is often slow to develop, short-lived, and context-dependent[3–5]. Therefore, considerable research has explored the neural circuits that govern these forms of learning. In the laboratory, Pavlovian fear conditioning and extinction procedures are widely used to study the neural basis of emotional memory. Briefly, animals learn an innocuous conditioned stimulus (CS) predicts an aversive unconditioned stimulus (US). After fear conditioning, animals exhibit conditioned fear responses (CRs), such as freezing, to presentation of the CS alone. Repeated presentation of the CS alone (i.e., extinction training) ultimately reduces conditioned responses[6,7]. Importantly, extinction represents new learning and does not erase the original fear memory. Fear to an extinguished CS returns under many circumstances, including when the CS is encountered outside of the extinction context, a phenomenon termed renewal[8,9]. Because extinction learning is at the heart of clinical interventions, such as exposure therapy, that are aimed to treat stress- or trauma-related disorders such as post-traumatic stress disorder (PTSD), many patients are prone to fear relapse[3,5].

Decades of research have implicated the hippocampus (HPC), medial prefrontal cortex (mPFC), and basolateral amygdala (BLA) in the encoding and context-dependent expression of extinction memories[2,10]. Recently, we have shown that the renewal of fear to an extinguished CS activates ventral hippocampal (vHPC) neurons projecting to both the mPFC and BLA[11–13]. Importantly, functional disconnection of the vHPC and either the prelimbic (PL) prefrontal cortex or BLA impairs fear renewal[12]. These studies support a circuit model in which vHPC projections to the mPFC and BLA facilitate the retrieval of CS-US memories when an extinguished CS is encountered outside the extinction context[12,13]. However, when the CS is encountered in the extinction context, the retrieval of fear memories must be suppressed in order to dampen fear responses, such as freezing, to the CS. Recent work in humans suggests that retrieval suppression might be mediated by prefrontal cortical projections to the hippocampus[14].

Anatomically, the mPFC does not project directly to the HPC, but it can influence the HPC through indirect projections. For example, the mPFC projects to midline thalamic nuclei that relay information to both the hippocampus and amygdala[15,16]. mPFC projections to the midline paraventricular nucleus, in particular, have been implicated in the expression of conditioned fear[17,18]. In addition, the nucleus reuniens (RE), a midline thalamic nucleus is well positioned to mediate mPFC influences on hippocampal function[16,19,20]. Lesions or inactivation of the RE impair forms of memory that require both the mPFC and HPC[21–23], including goal-directed spatial memory[24] and contextual fear memories[25,26]. Given the crucial role of the RE in mediating mPFC-HPC interactions, we sought to determine whether it also plays a role in the encoding and retrieval of context-dependent extinction memories. Using Pavlovian fear conditioning and extinction procedures in rats, we show that pharmacological inactivation of the RE dramatically increases freezing behavior during both the encoding and later retrieval of an extinction memory. This extinction impairment was not state-dependent. This pattern of extinction deficits was reproduced by selective pharmacogenetic silencing of mPFC neurons (or their terminals) projecting to the RE. Taken together, these data reveal a novel role for the prefrontal-reuniens circuit in the inhibition of fear after extinction. This circuit may function to oppose fear expression after threat has passed.

## Results

**RE inactivation impairs encoding of extinction.** To explore the role of RE in fear extinction, we first examined whether reversible inactivation of the RE with the $GABA_A$ agonist muscimol would impair the acquisition and later retrieval of the extinction memory. Because mPFC-HPC circuits have been implicated in contextual processing[2,27], we were particularly interested in whether RE inactivation might influence the context-dependence of the extinction memory. To this end, we examined the effects of RE inactivation on freezing during within-subject retrieval tests conducted in the extinction (ABB) and conditioning (ABA) contexts. Rats were first implanted with a single midline cannula targeting the RE (Fig. 1a, Supplementary Figure 1). After recovery from surgery, rats underwent fear conditioning, extinction, and retrieval testing (Fig. 1b). During fear conditioning (Fig. 1c, left), rats exhibited low levels of freezing behavior prior to the onset of the first conditioning trial, and an increase in freezing across the conditioning trials [repeated measures ANOVA, main effect of trial, $F_{(5, 115)} = 36.7$, $p < 0.001$]. The levels of freezing did not differ between the drug groups [$F < 1.8$], indicating that rats in each group acquired similar levels of conditioned fear. The following day, rats received intra-RE infusions of either saline (SAL) or muscimol (MUS) immediately prior to an extinction training session (45 CS-alone trials) that was conducted in a context different from that used for conditioning. During this session (Fig. 1c, middle), both groups of rats exhibited robust conditioned freezing to the CS in the earliest trial block, and saline-treated rats exhibited a within-session decrease in freezing that is typical of extinction learning. However, inactivation of the RE completely eliminated this within-session decrement in freezing [repeated measures ANOVA, main effect of drug, $F_{(1, 23)} = 14.86$, $p = 0.0008$; drug × trial interaction $F_{(9, 20)} = 6.46$, $p < 0.0001$].

Twenty-four hours after extinction, rats were tested for their fear to the extinguished CS in both the extinction (retrieval) and conditioning (renewal) contexts. As shown in Fig. 1c (right), rats extinguished under muscimol showed greater levels of CS-elicited freezing compared to control rats and this was particularly pronounced in the extinction context; both groups of rats renewed fear to the extinguished CS outside the extinction context. These observations were confirmed in a repeated measures ANOVA, which revealed main effects of drug [$F(1,2) = 5.14$; $p = 0.03$] and test context [$F_{(1,23)} = 16.85$; $p = 0.0004$]. Although there was not a reliable drug x test interaction [$F_{(1,23)} = 0.89$; $p = 0.36$], planned comparisons revealed a significant difference between SAL and MUS groups during the retrieval session ($p = 0.011$) but not in the renewal session ($p = 0.226$). These results reveal that RE inactivation causes a deficit in the acquisition of fear extinction.

**RE inactivation impairs extinction retrieval, but not fear renewal.** Given the critical role of the RE in encoding an extinction memory, we next examined the role of the RE in the retrieval those memories. Rats were implanted with a single midline cannula targeting the RE and, after recovery from surgery, underwent fear conditioning, extinction, and retrieval testing (see Fig. 2a for behavioral design). RE placements were similar to those in Supplementary Figure 1. Freezing behavior during the conditioning session is shown in Fig. 2b. As before, freezing was low before fear conditioning but significantly

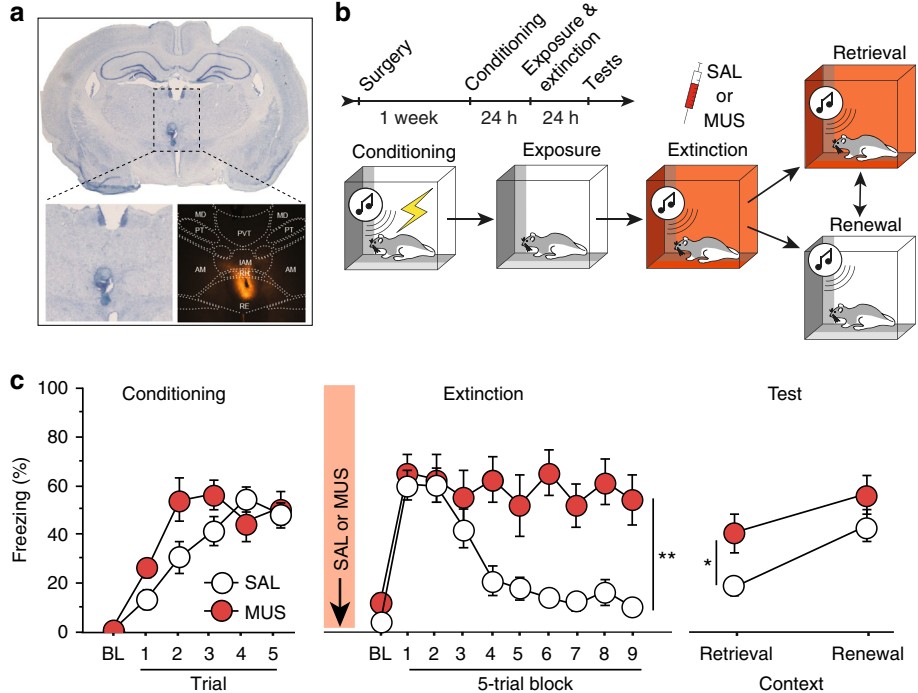

**Fig. 1** Muscimol inactivation of RE impairs encoding of fear extinction. **a** Representative thionin-stained coronal section showing cannula placement in the RE. The darkfield image shows diffusion of TMRx muscimol in the RE. **b** Schematized behavioral design. Illustrations are original artwork composed by the authors and adapted from ref[27]. **c** (Conditioning, left) Percentage of freezing during the 3-min baseline (BL) and 1-min interstimulus interval (ISI) following each CS-US pairing during the fear conditioning session. (Extinction, middle), Percentage of freezing during the 3-min baseline and 30-s ISIs across 9 extinction blocks (each block represents average freezing of 5 extinction trials) for the extinction session. Arrow indicates the timing of saline (SAL; white circles; n = 14) or muscimol (MUS; red circles; n = 11) infusion before the extinction session. (Retrieval tests, right), Average percentage of freezing for 5 CS test trials in the extinction (retrieval) and conditioning (renewal) contexts. All data are means ± s.e.m.s; *p < 0.05; **p < 0.01; one-way factorial and repeated measures ANOVA

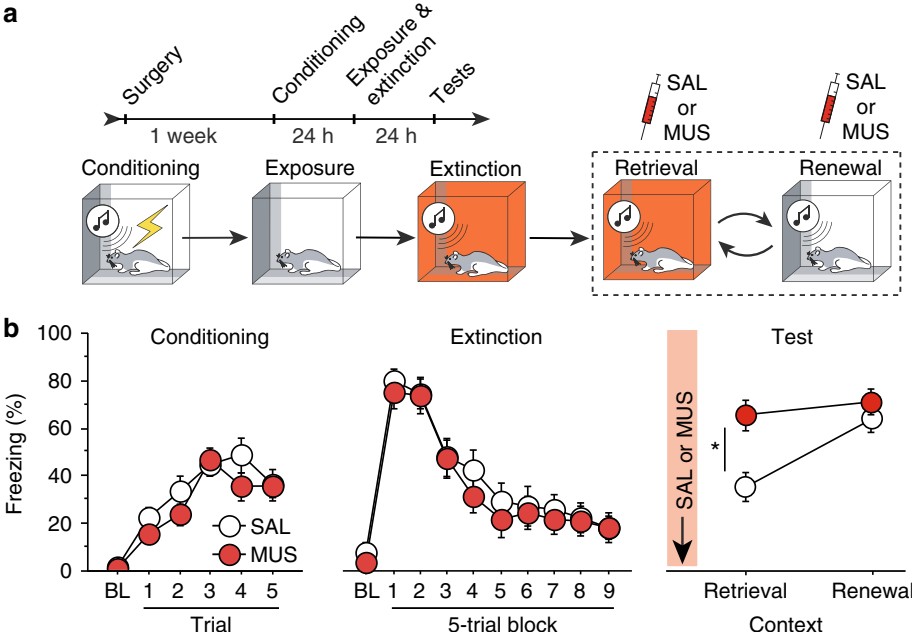

**Fig. 2** Muscimol inactivation of RE impairs the retrieval of fear extinction. **a** Schematized behavioral design. Illustrations are original artwork composed by the authors and adapted from ref [27]. **b** (Conditioning, left), Percentage of freezing during the 3-min BL and 1-min ISI following each CS-US pairing during fear conditioning. (Extinction, middle), Percentage of freezing during the 3-min BL and 30-s ISIs during extinction. (Test, right), Percentage of freezing averaged across 5 CS test trials in the extinction (retrieval) and conditioning (renewal) contexts (right). Arrows indicate the timing of saline (SAL; white circles; n = 16) or muscimol (MUS; red circles; n = 15) infusions before retrieval testing. All data are means ± s.e.m.s; * p < 0.05; ** p < 0.01; one-way factorial and repeated measures ANOVA

increased across the conditioning trials [repeated measures ANOVA, main effect of trial, $F_{(5, 145)} = 26.3$, $p < 0.0001$]. The following day the rats received extinction training in a different context. Rats showed high levels of CS-elicited freezing early in the session, but it dramatically decreased by the end of the session [repeated measures ANOVA, main effect of trial, $F_{(9, 261)} = 53.0$, $p < 0.0001$], indicating successful within-session extinction. There were no group differences observed during the conditioning and extinction sessions [$Fs < 1.6$].

On subsequent days, the rats received intra-RE infusions of SAL or MUS immediately prior to retrieval tests in the extinction and conditioning contexts; each rat was tested under the same drug condition for both tests. As shown in Fig. 2b, saline-treated rats exhibited significantly lower fear in the retrieval context relative to the renewal context. Importantly, MUS infusions into RE increased CS-elicited freezing in the extinction context, but not the renewal context. These observations were confirmed in a repeated measures ANOVA that revealed significant main effects of drug [$F_{(1,29)} = 8.79$; $p = 0.006$], test context [$F_{(1,18)} = 20.2$; $p = 0.0003$] and a significant drug × test interaction [$F_{(1,29)} = 4.43$; $p = 0.04$]. Post-hoc comparisons revealed a significant difference between SAL and MUS groups during the retrieval session ($p = 0.0024$) but not in the renewal session ($p = 0.28$). These results reveal that RE inactivation causes a deficit in the retrieval of extinction memories. Importantly, the increased freezing produced by RE inactivation was not due to nonspecific reductions in locomotor activity insofar as both pre-CS baseline freezing and fear renewal were unaffected by RE inactivation (Supplementary Fig. 2a). These results indicate that the RE is required for the inhibition of conditioned fear to an extinguished CS.

**Muscimol-induced extinction impairments are not state-dependent.** The previous results reveal that MUS infusions into the RE impair both the encoding and retrieval of fear extinction, but did not affect fear renewal. It is possible that this pattern of results is due to a shift in interoceptive (i.e., drug) context between extinction and retrieval testing that itself causes fear renewal[28]. To examine this possibility, we conducted an experiment in which RE inactivation occurred before both the extinction and retrieval sessions. If the interoceptive context associated with RE inactivation is critical for the expression of extinction, then animals that are extinguished and tested after RE inactivation should show normal extinction retrieval.

To examine this possibility, rats were implanted with a single midline cannula targeting the RE and after recovery from surgery underwent fear conditioning, extinction, and retrieval testing. Muscimol was infused in RE prior to both extinction and retrieval sessions. During the extinction session (Fig. 3, middle), we replicated our previous observation that RE inactivation impairs within-session extinction compared to saline controls [repeated measures ANOVA, main effect of group, $F_{(2,60)} = 12.8$; $p < 0.001$]. During retrieval testing (Fig. 3, right), animals extinguished and tested under RE inactivation continued to exhibit an extinction impairment relative to SAL-treated controls and exhibited levels of fear comparable to that in rats that did not undergo extinction. These observations were confirmed in an ANOVA performed on the average CS-elicited freezing during the test [main effect of group, $F_{(2,60)} = 4.8$; $p < 0.05$]. Post-hoc comparison revealed that SAL-treated rats differed from both MUS-treated and No-ext controls, which did not differ from one another. Importantly, these data indicate the extinction retrieval deficits in muscimol-treated rats are not due to a drug-shift induced renewal, because extinction deficits were observed in animals extinguished and

tested in the same drug state. These results indicate that encoding and retrieval deficits after MUS infusions into RE are not due to state-dependent generalization deficits.

Another possibility is that extinguishing the animals outside the extinction context more strongly contextualizes the extinction memory than delivering extinction trials in the conditioning context[29]. This might increase the sensitivity of extinction to RE inactivation. To examine this issue, we compared the effects of RE inactivation on extinction retrieval in rats that underwent extinction in either the conditioning context (COND) or a novel context (NOVEL); all rats were then tested in their respective extinction contexts (AAA or ABB) and then in a novel renewal context (C) (see Supplementary Figure 3a for behavioral paradigm). Animals were first implanted with cannulas targeting RE and, after recovery from surgery, underwent fear conditioning in context A (Supplementary Figure 3b). On Day 2, animals were extinguished in either the conditioning context (COND) or a novel context (NOVEL). During the extinction session, rats showed high levels of CS-elicited freezing early in the session, but it dramatically decreased by the end of the session indicating successful extinction [repeated measures ANOVA, main effect of trial $F_{(1,11)} = 13.07$; $p = 0.0041$]. Freezing in rats extinguished in the conditioning context was significantly higher than that in rats extinguished in the novel context [repeated measures ANOVA, main effect of extinction context $F_{(1,11)} = 11.24$; $p = 0.007$], which reflects a summation of context and CS fear in the conditioning context.

On subsequent days, animals received infusions of SAL or MUS (counterbalanced, within-S's design) and a retrieval test in the extinction context followed by a test in a third novel context (renewal test). During the retrieval test (Supplementary Figure 3b, right), MUS infusions into the RE increased freezing to the extinguished CS independent of the extinction procedure; MUS infusions did not affect the renewal of freezing outside the extinction context. These observations were confirmed in a one-way repeated-measures ANOVA that revealed a main effect of drug [$F_{(1,11)} = 37.57$; $p < 0.001$], but no effect of extinction context [$F_{(1,11)} = 0.79$; $p = 0.39$] or drug × context interaction [$F_{(1,11)} = 2.44$; $p = 0.15$]. During the renewal session, a one-way repeated measures ANOVA revealed no main effect of drug [$F_{(1,11)} = 2.98$; $p = 0.12$] or extinction context [$F_{(1,11)} = 0.21$; $p = 0.65$] and no interaction between the two variables $F_{(1,11)} = 2.13$; $p = 0.13$].

**Encoding and retrieval of extinction increases Fos expression in RE.** The previous results indicate that RE inactivation impairs both the encoding and retrieval of extinction memories. Here we sought to determine whether RE neurons are activated (as indexed by c-Fos immunohistochemistry) during the encoding and retrieval of extinction memories. To this end, we examined Fos expression in the RE after the extinction training session, as well as after extinction retrieval. In the first case (Fig. 4a), rats underwent auditory fear conditioning followed 24 h later by extinction training; the animals were sacrificed 90 min after the end of the extinction session. Conditioning and extinction of fear were similar to previous experiments (Fig. 4c). As shown in Fig. 4b–d, animals that underwent fear extinction exhibited significantly higher number of Fos+ neurons in the RE compared to home control rats [unpaired t test, $t_{(12)} = 5.5$, $p < 0.001$]. Retrieval testing also increased Fos expression in the RE. As shown in Fig. 4e, after conditioning and extinction, conditioned freezing to the extinguished CS was suppressed in the retrieval context and elevated in the renewal context. Interestingly, testing in either the retrieval or renewal contexts increased the number of Fos+ neurons in RE relative to home-cage controls (Fig. 4f). A one-way

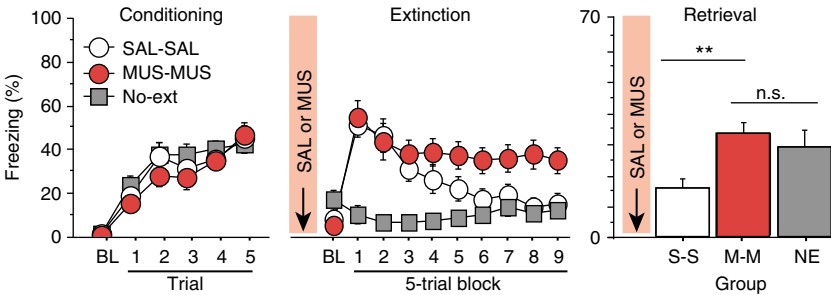

**Fig. 3** Muscimol-induced extinction impairments are not state-dependent. (Conditioning, left), Percentage of freezing during the 3-min BL and 1-min ISI following each CS-US pairing during the fear conditioning session. (Extinction, middle), Percentage of freezing during the 3-min BL and 30-s ISIs across the extinction session. (Retrieval, right), Percentage of freezing for the 5 CS test trials in the extinction (retrieval) context. Arrows indicate the timing of muscimol (MUS; red circles; n = 20) or saline (SAL; white circles; n = 20) infusions into RE before either extinction training or retrieval testing. No-ext rats (gray squares; n = 23) were placed in the extinction context during but did not receive CS presentations. All data are means ± s.e.m.s; *p < 0.05; **p < 0.01; one-way factorial and repeated measures ANOVA

ANOVA revealed a main effect of group [$F_{(2,25)} = 3.8$, $p < 0.05$] and post-hoc comparisons revealed that both SAME and DIFF groups were reliably higher than the HOME control and did not differ from one another. These results reveal that RE neurons are recruited during both encoding and retrieval of extinction memories (including extinction memories undergoing renewal).

**Extinguished CSs increase single-unit firing in the RE.** The previous data indicate that extinguished CSs increase Fos expression in the RE in both the extinction and renewal contexts. However, Fos expression has low temporal resolution and integrates neuronal activity elicited by both the context and CS during retrieval testing. It is therefore possible that RE neurons respond differentially to CSs presented in the extinction and renewal contexts. To examine this possibility, we made single-unit recordings from RE neurons in freely behaving rats using a within-subject design. A schematic illustration of the behavioral paradigm is shown in Fig. 5a. Briefly, animals were implanted with a microwire bundle targeting RE (see Fig. 5b for representative electrode placements). After recovery from surgery, animals underwent auditory fear conditioning followed 24 h later by extinction training.

Twenty-four hours after extinction, the rats received an unsignaled reminder shock in context A to facilitate the return of freezing during the renewal test. On the subsequent day, rats were subjected to a within-subject testing procedure wherein the extinguished CS was presented in both the extinction context (retrieval) and a novel context (renewal); single-unit recordings were made during both tests and the same neurons were tracked across sessions. During the retrieval tests (Fig. 5d), rats showed lower levels of freezing in the extinction context relative to the renewal context, though this was not statistically reliable [$F_{(1,2)} = 17.10$; $p = 0.053$ for trial 1]. During the retrieval tests we recorded from a total of 27 neurons in RE. The basal firing rate of these neurons was significantly higher in the retrieval (2.88 ± 0.17 Hz) than the renewal (2.42 ± 0.22 Hz) [paired t test; $t_{(26)} = -2.3$, $p < 0.03$]. Among this population of cells, seven neurons (25%) exhibited significant increases in firing to the tone CS (defined as an increase in firing rate > 1.96 standard deviations above the 500 ms pre-CS baseline). Interestingly, tone-responsive RE neurons exhibited greater CS-evoked firing within 200 ms of CS onset in the extinction context relative to that in the renewal context (Fig. 5c, d). This observation was confirmed in a one-way repeated measures ANOVA that revealed a main effect of test [$F_{(1,6)} = 15.67$; $p = 0.008$] indicating that neurons in RE showed greater CS-evoked responses to an extinguished CS in the extinction context relative to the renewal context.

**Silencing RE projectors in the mPFC impairs extinction encoding.** The mPFC plays a critical role in extinction learning and retrieval. The RE receives heavy input from the mPFC and this may represent a critical functional input regulating fear extinction. Here we sought to determine whether mPFC projections to the RE are involved in the acquisition and retrieval of fear extinction. Rats received injections of AAV5-Cre-GFP in the RE and AAV8-hSyn-DIO-hM4D(Gi)-mCherry in the mPFC 4–5 weeks prior to behavioral training (see Fig. 1b for behavioral design and Fig. 6a for viral expression). Twenty-four hours after auditory fear conditioning [repeated measures ANOVA, main effect of trial, $F_{(5,160)} = 35.6$; $p < 0.001$] (Fig. 6b, left), rats received systemic injections of either SAL or CNO and underwent fear extinction. As shown in Fig. 6b (middle), CNO administration increased CS-elicited freezing during the extinction session [repeated measures ANOVA, main effect of drug $F_{(1,32)} = 4.15$; $p = 0.05$.

During retrieval testing (Fig. 6b, right), all animals exhibited low levels of freezing in the extinction context and increased freezing to the CS in the renewal context [repeated measures ANOVA, main effect of test $F_{(1,32)} = 17.57$; $p = 0.0002$]. Interestingly, rats that previously received CNO during extinction training showed higher levels of freezing compared to SAL-treated rats during both of the retrieval tests [repeated measures ANOVA, main effect of drug, $F_{(1,31)} = 8.23$; $p = 0.007$]. Post-hoc comparisons revealed that CNO-injected animals showed elevated levels of freezing compared to SAL-injected animals during both retrieval ($p = 0.031$) and renewal ($p = 0.011$) sessions. These results are consistent with the effects that we previously showed with RE inactivation alone and reveal that projections from the mPFC to the RE are involved in extinction learning. Furthermore, this effect was not simply a performance effect of CNO (e.g., non-specific increases in freezing), insofar as pre-CS baseline freezing during extinction training was not affected by CNO [unpaired t test; $t_{(32)} = 0.18$; $p = 0.85$] and the extinction impairments were manifest during the drug-free retrieval tests.

**Silencing RE projectors in the mPFC impairs extinction retrieval.** Next, we examined whether mPFC projections to RE also mediate the retrieval of extinction memories. Rats received injections of CAV2-Cre in the RE and AAV8-hSyn-DIO-hM4D(Gi)-mCherry in the mPFC 4–5 weeks prior to behavioral training (See Fig. 2a for behavioral design). As shown in Fig. 6c, rats underwent auditory fear conditioning [repeated measures ANOVA, main effect of trial, $F_{(5,30)} = 9.0$; $p < 0.001$] and three sessions of extinction [repeated measures ANOVA, main effect of session, $F_{(2,12)} = 16.0$; $p < 0.001$]. On the following two days after the last extinction session, rats received extinction retrieval tests

 

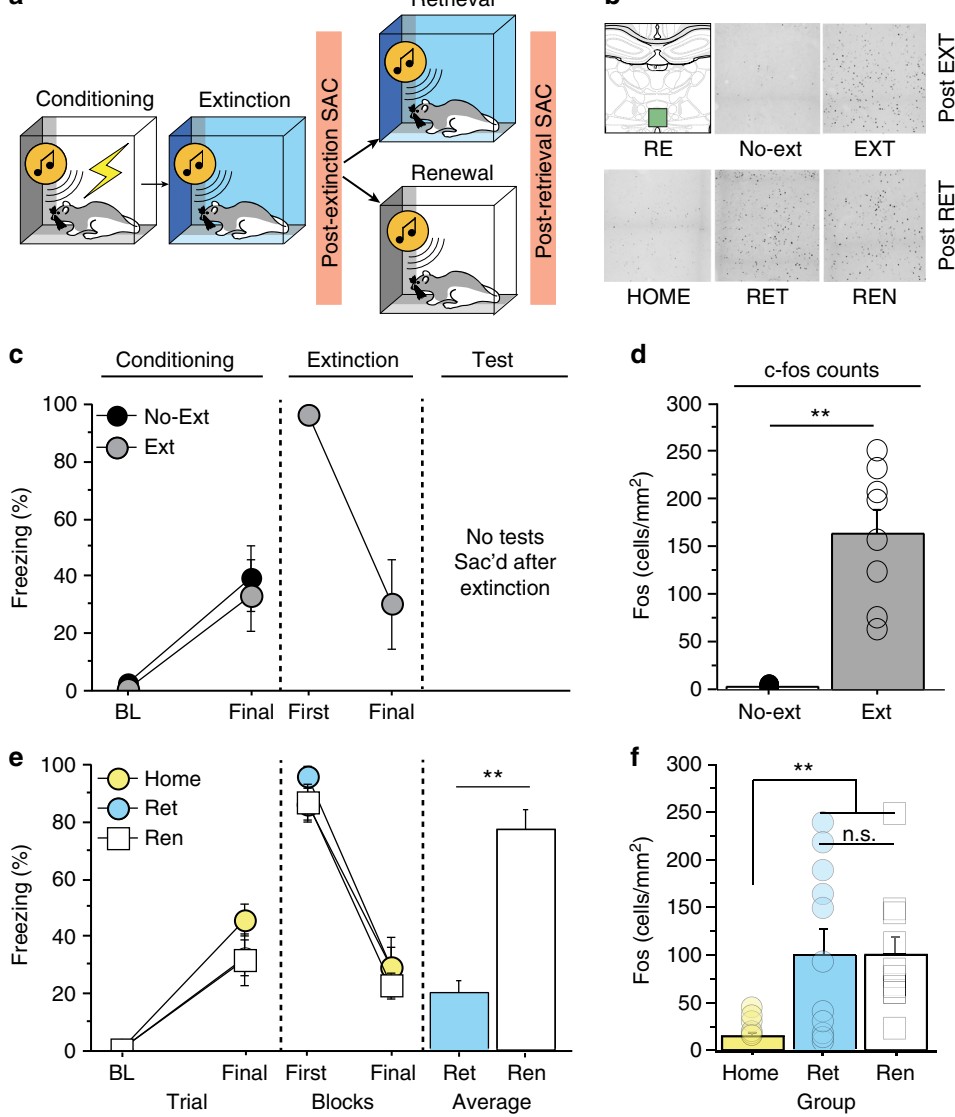

**Fig. 4** Extinction encoding and retrieval increases c-fos expression in the RE. **a** Schematized behavioral design. Illustrations are original artwork composed by the authors and adapted from ref [27]. **b** Representative coronal sections showing c-fos labeling in each group. **c** (Conditioning, left), Percentage of freezing during the 3-min baseline (BL) and 1-min interstimulus interval (ISI) after the last CS-US pairing during fear conditioning. (Extinction, middle), percentage of freezing during the first and last extinction blocks (each block represents average freezing of 5 ISIs) for the extinction training session (black circles, no-ext; open circles, extinction). **d** Number of c-fos-positive neurons (per mm²) in RE of rats that received extinction (EXT; $n = 8$) or no extinction (No-ext; $n = 6$). **e** Percentage of freezing during fear conditioning (left), extinction (middle; each block represents average freezing of 5 ISIs for the first and last session), and the retrieval test (right). The tests show average freezing for 5 CS test trials in the extinction (retrieval; blue circle) and conditioning (renewal, white square) contexts. **f** Number of c-fos-positive neurons (per mm²) in the RE in rats that received an extinction retrieval test (RET; $n = 11$), fear renewal test (REN; $n = 11$) and home control (Home; $n = 6$). All the data are means ± s.e.m.s; *$p < 0.05$; ** $p < 0.01$; one-way factorial and repeated measures ANOVA

using a within-subjects design in which each animal served as its own control. That is, rats were tested after receiving either SAL or CNO in two counterbalanced tests in the extinction context, which were conducted over two days. As shown in Fig. 6c (right), CNO administration impaired the retrieval of the extinction memory and increased freezing in the extinction context [repeated measures ANOVA, main effect of drug, $F_{(1,6)} = 7.2$; $p < 0.05$]. Importantly, CNO did not increase baseline freezing prior to CS onset [repeated measures ANOVA, no main effect of drug $F_{(1,6)} = 0.55$; $p = 0.49$] (Supplementary Fig. 2b). These results indicate that prefrontal projections to the RE are involved in both the encoding and retrieval of extinction memory.

**Silencing mPFC terminals in RE impairs extinction retrieval**. The previous experiment reveals that mPFC neurons that project to RE are involved in both the encoding and retrieval of extinction memories. However, silencing these projection neurons might influence mPFC output to other brain areas insofar it has been shown that mPFC neurons send collateral projections to medio-dorsal thalamus and reticular thalamus[30]. To specifically manipulate mPFC projections to RE, we expressed inhibitory DREADDs (or a blank control) in the mPFC and microinfused CNO into the RE to inactivate mPFC terminals[31]. A schematic illustration of the behavioral paradigm is shown in Fig. 7a. Rats received injections of AAV8-hSyn-hM4D(G$_i$)-mCherry or

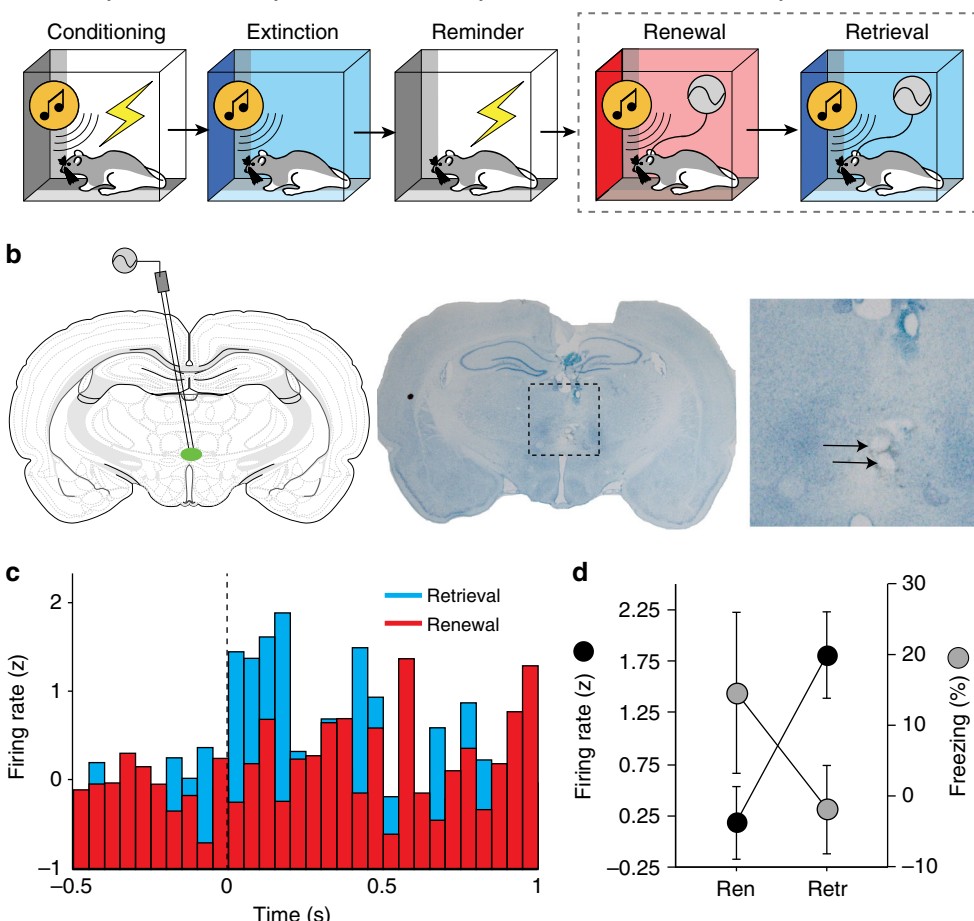

**Fig. 5** Extinction retrieval increases CS-elicited spike firing in the RE. **a** Schematic behavioral design. Illustrations are original artwork composed by the authors and adapted from ref [27]. **b** Representative coronal sections showing electrode placements in RE. Illustrations are original artwork adapted from open access brain atlas templates[44]. **c** Average normalized firing rate among RE neurons ($n = 7$) across five CS presentations in either the retrieval (blue bars) or renewal context (red bars). Firing rate was binned (50 ms) during a 500 ms pre-CS period and a 1-sec post-CS period. **d** Average firing rate of RE neurons over 5 trials during the first 200 ms of tone onset (black circles) and percentage of freezing for the 5 trials during testing in retrieval and renewal context (gray circles). All data are means ± s.e.m.s; * $p < 0.05$; ** $p < 0.01$; repeated measures ANOVA

AAV8-hSyn-GFP in either PL or IL (Fig. 7b) or both and were implanted with cannula targeting the RE five weeks after viral infusions. Viral infusions in the mPFC produced robust terminal expression in the RE (Supplementary Figure 4).

One week after cannula implantation, rats underwent auditory fear conditioning [repeated measures ANOVA, main effect of trial, $F_{(5,240)} = 50.72$; $p < 0.001$, no main effect of group $F_{(3,48)} = 0.604$, $p = 0.61$] and three sessions of extinction training [repeated measures ANOVA, main effect of trials, $F_{(1,48)} = 132.45$; $p < 0.0001$, no main effect of group $F_{(3,48)} = 0.28$; $p = 0.84$] (Supplementary Figure 4). On the following two days after the last extinction session, rats received extinction retrieval tests using a within-subjects design in which each animal served as its own control (Fig. 7c). That is, rats were tested after receiving infusions of either SAL or CNO in two counterbalanced tests in the extinction context, which were conducted over two days. As shown in Fig. 7c, CNO infusion into RE increased conditional freezing to the extinguished CS in animals expressing inhibitory DREADDs in either the PL, IL or both areas; CNO did not affect freezing in blank controls. These observations were confirmed in a repeated measures ANOVA that revealed a main effect of drug [$F_{(1,48)} = 17.74$; $p = 0.0001$], without a main effect of group, [$F_{(3,48)} = 1.73$; $p = 0.17$] or a group x drug interaction [$F(3,48) = $

2.02; $p = 0.12$]. Planned comparisons revealed that CNO infusions did not result in any changes in freezing in rats receiving blank GFP virus ($p = 0.74$) confirming that CNO-induced increases in freezing are not due to nonspecific effects of the drug. However, CNO infusions increased freezing in all three groups expressing inhibitory DREADDs in the mPFC: PL + IL ($p = 0.022$), PL ($p = 0.012$), and IL ($p = 0.046$). These results indicate that both prelimbic and infralimbic prefrontal projections to the RE are involved in the retrieval of an extinguished fear memory.

## Discussion
Here we have demonstrated for the first time that the nucleus reuniens of the midline thalamus is required for both encoding and retrieving extinction memories. Extinction training or retrieval testing increased the activity of RE neurons and inactivation of the RE or its projections from the mPFC produced deficits in extinction memory. Taken together, the present study reveals a novel role for prefrontal-thalamic circuits in fear extinction and suggests the RE is a key structure mediating prefrontal top-down inhibitory control of fear inhibition that is crucial for extinction.

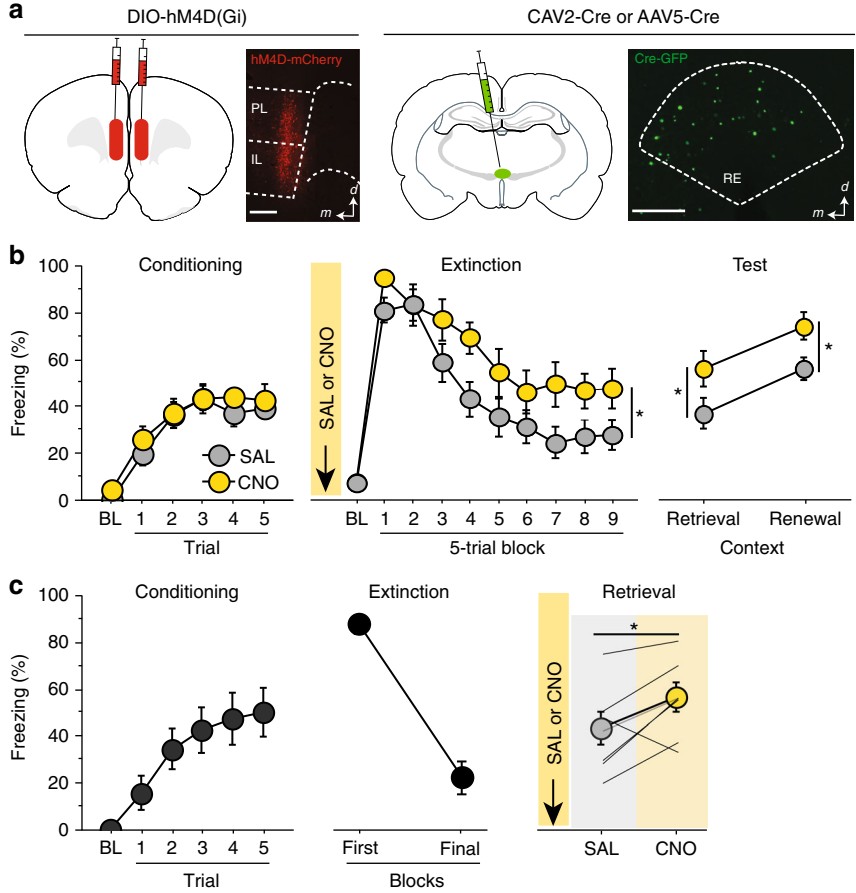

**Fig. 6** Silencing mPFC→RE projectors impairs encoding and expression of fear extinction. **a** Representative images of Cre-dependent DREADD expression; AAV8-hSyn-DIO-hM4D(G$_i$)-mCherry in the mPFC and AAV5-Cre-eGFP virus in the RE. Illustrations are original artwork adapted from open access brain atlas templates[44]. **b** (Conditioning, left), Percentage of freezing during the 3-min baseline (BL) and 1-min interstimulus interval (ISI) following each CS-US pairing during the fear conditioning session. (Extinction, middle), Percentage of freezing during the 3-min baseline and 30-s ISIs across 9 extinction blocks (each block represents average freezing of 5 extinction trials) for the extinction training session. (Test, right), Percentage of freezing for 5 CS test trials in the extinction (retrieval) and conditioning (renewal) context. The arrow indicates the timing of the CNO (CNO; yellow circles; $n = 15$) or saline (SAL; white circles; $n = 19$) injections before the extinction session. **c** (Conditioning, left), Percentage of freezing during the 3-min baseline (BL) and 1-min interstimulus interval (ISI) following each CS-US pairing during the fear conditioning session. (Extinction, middle), Percentage of freezing during the first block of three extinction sessions (each block represents average freezing of 5 ISIs). (Retrieval, right), Average percentage freezing during 5 CS test trials during extinction retention tests after either SAL or CNO injection. Shaded panel shows average ISI freezing across 5 ISIs and each gray line represents an individual rat ($n = 7$). Scale bar represent 0.5 mm for mPFC and 0.25 mm for RE histology images. All the data are means ± s.e.m.s; *$p < 0.05$; **$p < 0.01$; repeated measures ANOVA

The fact that the RE is critically involved in extinction learning and recall is in line with previous work demonstrating the importance of the midline thalamus in both memory and emotion[23–25,32–35]. Importantly, a recent study demonstrated that the RE is important for maintaining the specificity of contextual fear memory[25]. Specifically, the authors showed that RE inactivation caused an overgeneralization of conditional fear to contexts other than the one in which shock was encountered, but did not affect fear recall in the original conditioning context or auditory fear expression[25]. Interestingly, both contextual conditioning and the extinction of fear to an auditory CS rely heavily on contextual processing. That is, contextual fear conditioning requires the acquisition of a contextual representation that comes in to association with an aversive US. As a result, conditioned fear is expressed in the place where shock is encountered, but not in other places. Similarly, extinction involves learning that a CS is not reinforced in a particular context. In this case, the suppression of fear to a CS after extinction occurs in the extinction context, but not in other places; in other words, fear to an

extinguished CS renews outside the extinction context[2]. In both cases, deficits in contextual specificity—knowing what happened where—would result in both overgeneralized fear after context conditioning and an inappropriate renewal of fear after extinction. In both cases, fear is expressed in otherwise safe contexts. Together, these data suggest that the RE and its connections with the mPFC might be involved in the inhibition of fear in safe contexts. Importantly, RE or mPFC-RE projections were not involved in mediating the renewal of fear to an extinguished CS outside the extinction context.

Previous studies from our lab have demonstrated that the renewal of extinguished fear requires the hippocampus and its projections to the mPFC[11–13,27]. HPC inactivation or disconnections of the HPC and mPFC disrupted fear renewal, but did not affect the expression of extinction[12,36]. This reveals that direct HPC-mPFC projections are not involved in fear inhibition, but rather contribute to the excitation of fear to an extinguished CS outside the extinction context. In the present study, we have shown that direct mPFC inputs to the RE are crucial for fear

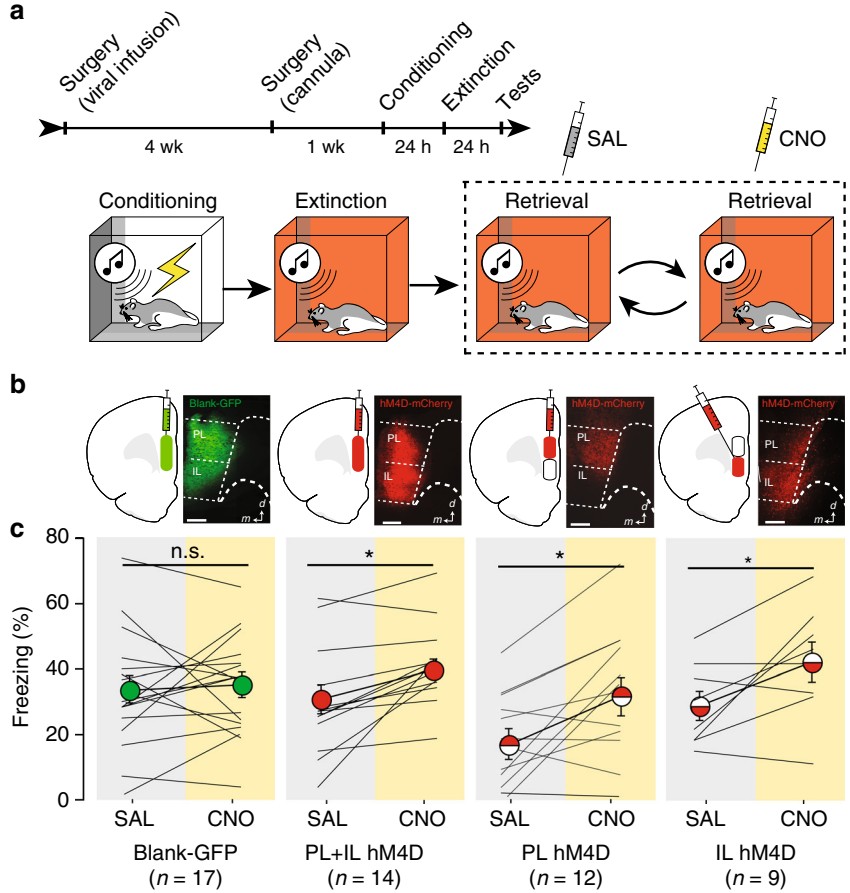

**Fig. 7** Silencing mPFC terminals in RE impairs extinction retrieval. **a** Schematic behavioral design. Illustrations are original artwork composed by the authors and adapted from ref [27]. **b** Representative images of viral expression in the mPFC. Illustrations were adapted from open access brain atlas templates[44]. **c** Average percentage of freezing during the retrieval tests. Freezing during each test is averaged across five CS test trials either SAL or CNO infusions in RE. Scale bars represent 0.5 mm. All the data are means ± s.e.m.s; *$p < 0.05$; **$p < 0.01$; repeated measures ANOVA

extinction. Indeed, the RE has been suggested as a critical hub that interconnects the mPFC and hippocampus[16,19]. Prefrontal projections to the RE are involved in fear memory generalization[25], goal-directed spatial navigation[24], motivation and reward related behavior[37] and spatial working memory[19,23]. Anatomically, there are strong reciprocal projections between the RE and the mPFC and hippocampus[38], and the RE is important for synchronizing local field potentials in this circuit[39]. This suggests that the mPFC-RE interactions we find are important for extinction learning might ultimately be mediated through RE projections to the hippocampus. Indeed, interactions between the mPFC and HPC are involved in a number of emotional and cognitive processes associated with fear and anxiety including extinction[2,9,11,20,27,40–42].

Indeed, a role for mPFC-RE-HPC circuits in the inhibition of conditioned fear in safe contexts might be an example of a broader role of this circuit in *retrieval suppression*. For example, humans can actively suppress recalling a particular memory, either by being instructed to do so in the laboratory or spontaneously when confronted with a reminder of a trauma. Interestingly, functional neuroimaging work indicates that retrieval suppression is associated with an increase in activity in the mPFC, but a suppression of activity in the hippocampus[14]. It has been suggested that RE might coordinate this inhibitory influence of the mPFC on hippocampal memory retrieval. In the context of the present work, this mechanism might eliminate interference between fear and extinction memories, by suppressing the

retrieval of the fear memory in otherwise safe contexts. That is, during extinction retrieval, when an extinguished CS is encountered in a safe context, a retrieval suppression process mediated by projections of the mPFC to the HPC via the RE might prevent retrieval of the fear memory. Alternatively, RE projections to the amygdala, including the basolateral and basomedial amygdala[43], might allow for both the mPFC and HPC to exert integrated contextual control over the expression of fear.

In conclusion, the present study reveals that mPFC inputs to the RE are critically involved in fear extinction. Given the critical role for mPFC-HPC interactions in emotional memory retrieval, we propose that the RE serves as a hub by which mPFC regulates HPC activity to suppress the retrieval of fear memories. Inhibition of either the RE or its inputs from the mPFC impairs both extinction encoding and retrieval, resulting in excessive fear in an otherwise safe context. Preventing fear relapse is at the core of exposure therapy; therefore, future studies are needed to understand how dysfunction in mPFC-RE circuits underlies psychopathology associated with stress- and trauma-related events.

## Methods

**Subjects**. Adult male rats (200–224 g; Long-Evans Blue Spruce) obtained from Envigo were used for the experiments. The rats were individually housed on a 14/10 h light/dark cycle and had access to food and water *ad libitum*. All experiments were performed during the light cycle. The rats were handled for 30 s every day for 5 days before the experiments to habituate them to the experimenters. All experimental procedures were performed in accordance with the protocols approved by the Texas A&M University Animal Care and Use Committee.

**Viruses and drugs**. AAV8-hSyn-DIO-hM4D(G$_i$)-mCherry (titer ≥ 4 × 10$^{12}$ vg/mL) was obtained from University of North Carolina Vector Core and Addgene. CAV2-Cre (titer: 8.7 × 10$^{12}$ pp/mL) was obtained from the Institute of Molecular Genetics of Montpellier and AAV5-CMV-HI-eGFP-Cre-WPRE-SV40 (titer: 0.64–1.42 × 10$^{14}$ GC/mL) was from University of Pennsylvania Vector Core. AAV-8-hsyn-hM4D(Gi)-mCherry (titer: 3 × 10$^{12}$ vp/mL) and AAv8-hSyn-eGFP (titer: 3 × 10$^{12}$ vp/mL) was obtained from Addgene. Clozapine-N-oxide (CNO) was provided by the National Institute of Mental Health (NIMH; Chemical Synthesis and Drug Supply Program) and muscimol (GABA$_A$ receptor agonist) was from Sigma.

**Surgery**. For muscimol microinfusion experiments, rats were anesthetized with isoflurane (5% for induction, ~2% for maintenance), and placed into a stereotaxic instrument (Kopf Instruments). An incision was made in the scalp, the head was leveled, and bregma coordinates were identified. Small holes were drilled in the skull to affix three jeweler's screws and to target a single midline cannula (8 mm, 26 gauge; Plastics One) above the RE. The cannula was implanted at a 10° angle on the midline (A/P: −2.05 to 2.15 mm, M/L: + 1.0 mm, D/V: −6.7 to 6.9 mm from dura; coordinates were measured from bregma). The cannula was affixed to the skull with dental cement, and a stainless-steel dummy cannula (30 gauge, 9 mm; Plastics One) was inserted into the guide cannula. Rats were allowed to recover for a period of 7 d after surgery before behavioral testing.

For DREADD experiments targeting the mPFC→RE circuit, rats were bilaterally infused with AAV8-hSyn-DIO-hM4D(G$_i$)-mCherry into the mPFC (including PL and IL), and CAV2-Cre or AAV5-Cre-eGFP into the RE. Within the mPFC, two infusions (1.0 µl each) were made in the IL (A/P: +2.7–3.0 mm, M/L: ±0.5–0.75 mm, D/V: −4.4 mm from dura) and PL (A/P: + 2.7–3.0 mm, M/L: ± 0.75 mm, D/V: −3.2 mm from dura), respectively. A single infusion (1.0–1.2 µl) was made in the RE (A/P: −2.15 mm, M/L: +1.0 mm, D/V: −6.9 mm from dura) at a 10° angle.

For the terminal inactivation experiment, rats were bilaterally infused with AAV8-hSyn-eGFP into PL and IL using the coordinates mentioned above. For the active virus groups, targeting of PL and PL + IL groups was done using the coordinates mentioned above. However, for the IL group the following coordinate was used in order to limit the damage to PL (A/P = +2.7–3.0, M/L = ± 2.0, D/V = −4.9 from dura at 30° angle).

**Drug delivery**. For RE microinfusions, rats were transported to an infusion room using white buckets (5-gallon) from the vivarium. The dummy cannula was removed from the implanted guide and a stainless steel injector (33 gauge, 9 mm) connected to tubing was inserted into the guide cannula for intracranial infusions. Polyethylene tubing connected the injectors to Hamilton syringes (10 µl), which were mounted in an infusion pump (Kd Scientific). Infusions were monitored by the movement of an air bubble that separated the drug or saline solutions from distilled water within the polyethylene tubing. All infusions were made ~10 min before either extinction training or retrieval sessions. Muscimol was diluted in sterile saline to a concentration of 0.1 µg/µl. For terminal inactivation experiment, CNO dissolved in SAL (with 2.5% DMSO) at 1 mM concentration. Infusions were made at a rate of 0.1 µl/min for 3 min (0.3 µl total) and the injectors were left in place for 1 min for diffusion. After infusions, clean dummies were secured to the guide cannula.

For DREADD experiments, CNO was first dissolved in 2.5% DMSO and then diluted in sterile saline (0.9%) to a concentration of 3 mg/ml immediately before injection. Approximately 30–40 min before extinction or testing session, rats received intraperitoneal injection of either CNO (3 mg/kg) or saline in the vivarium and then were placed back to their home cages until the start of the behavioral procedures.

**Behavioral apparatus and contexts**. Sixteen identical rodent conditioning chambers (30 × 24 × 21 cm; Med-Associates, St Albans, VT) were used in all behavioral sessions. Each chamber consisted of two aluminum sidewalls and a Plexiglas ceiling and rear wall, and a hinged Plexiglas door. The floor consisted of 19 stainless steel rods that were wired to a shock source and a solid-state grid scrambler (Med-Associates) for the delivery of footshocks. A speaker mounted on the outside of the grating in one aluminum wall was used to deliver auditory stimuli. Additionally, ventilation fans and house lights were installed in each chamber to allow for the manipulation of contexts. Each conditioning chamber rest on a load-cell platform that is used to record chamber displacement in response to each rat's motor activity and is acquired online via Threshold Activity software (Med-Associates). For each chamber, load-cell voltages are digitized at 5 Hz, yielding one observation every 200 ms. Freezing was quantified by computing the number of observations for each rat that had a value less than the freezing threshold (load-cell activity = 10). Freezing was only scored if the rat is immobile for at least 1 s. Stimuli were adjusted within conditioning chambers to generate two distinct contexts in two distinct behavioral rooms. For context A, a 15-W house light was turned on, and the room light remained on. Ventilation fans (65 dB) were turned on, cabinet doors were left open, and the chambers were cleaned with 1% ammonium hydroxide. Rats were transported to context A in white plastic boxes. For context B, house lights were turned off and a fluorescent red room light was

turned on. The cabinet doors were closed and the chambers were cleaned with 1–1.5% acetic acid. Rats were transported to context B in black plastic boxes.

**Behavioral procedures**. For muscimol inactivation experiments, ~1 week after surgery, rats underwent fear conditioning, extinction, and retrieval testing in either the conditioning context (Context A) or the extinction context (Context B). Auditory fear conditioning consisted of five tone (CS; 10 s, 80 dB, 2 kHz)-footshock (US; 1.0 mA, 2 s) pairings with 60 s interstimulus intervals (ISIs). On the following day, rats underwent fear extinction in which they received a 3 min stimulus-free BL followed by 45 tone-alone presentations (30 s ISIs). Prior to the extinction session, rats were exposed to the conditioning context for 35 min 30 s to extinguish fear associated with the context. On the following two days, rats received a retrieval test in the conditioning context to assess fear renewal and a subsequent test in the extinction context to assess extinction retrieval. Each test consisted of a 10-min stimulus-free baseline period followed by 5 CS presentations (30 s ISIs). Rats received microinfusions of SAL or MUS into the RE 10-min before extinction training or retrieval testing. The test order was counterbalanced such that half of the rats received the renewal test first and the others received the retrieval test first.

To assess the state-dependence of RE inactivation, rats underwent fear conditioning, extinction and extinction retrieval testing as previously described. One group of rats received MUS infusions in the RE before both extinction training and the retrieval test and a second group of rats received SAL infusions before both extinction and retrieval test. A third group of rats (No-ext) also received SAL infusions, but they did not receive CS presentations during the extinction session. To assess whether extinction retrieval is affected in same context, the animals were conditioned and extinguished as described above. On the following 2 days, rats received either infusions of SAL or MUS (counterbalanced across days) prior to the retrieval test in the extinguished context to test the strength of the extinction memory. On the subsequent two days, rats received either infusions of SAL or MUS (counterbalanced across days) prior to the retrieval test in a novel non-extinguished context to test for fear renewal.

To determine whether extinction training or retrieval testing activates the RE, we examined Fos expression in the RE after these procedures. Rats underwent fear conditioning and extinction as previously described (though in this experiment the animals received three days of extinction training because they exhibited particularly high levels of freezing). One group of rats (EXT) was sacrificed and perfused 90 min after the first extinction session. Control rats (No-ext) stayed in their home cage during first extinction and were sacrificed together with EXT rats. After all extinction sessions, one group of rats underwent a renewal test (context A) and a second group received an extinction retrieval test (context B); a third group of rats (HOME) remained in their home cages during behavioral testing. Ninety-minutes after testing, all rats were sacrificed and perfused for c-fos analysis.

For the intersectional DREADD experiments, rats underwent auditory fear conditioning, extinction, retrieval testing 4–5 weeks after surgery as previously described. Rats received SAL or CNO injections either 30 min before extinction training or retrieval testing. For the encoding experiment, retrieval tests were conducted in both the conditioning context (context A, renewal) and the extinction context (context B, retrieval). For the retrieval, experiment animals were only tested in the extinction context (context B, retrieval) using a within-subjects procedure in which each animal served as its own control. That is, each rat received either a SAL or CNO injection before each of two extinction retrieval tests conducted over two days; test order was counterbalanced such that half of the animals received SAL in their first test whereas the other half received CNO in their first test.

For the terminal DREADD experiment, animals underwent surgery for viral infusions into either PL or IL or both. Five weeks after this surgery, animals underwent a second surgery to implant cannula targeting the RE. One week after the second surgery, rats underwent auditory fear conditioning, extinction, retrieval testing as previously described. Rats received infusions of SAL or CNO in RE 10 min before the retrieval testing using a within-subjects procedure in which each animal served as its own control.

**Electrophysiological recordings**. For the in-vivo electrophysiological recording experiment, a modified rodent conditioning chamber (30 × 24 × 21 cm) was used for the extinction and testing sessions. This chamber was modified to allow for awake, behaving recordings. One week after recovery from surgery, rats (n = 3) underwent auditory fear conditioning in context A in which they were presented with 3 CS-US (60 s ISI) pairings after a 3 min stimulus-free baseline period. On the subsequent two days, rats underwent identical extinction sessions in context B in which they were presented with 45 CS-alone trials (30 sec ISI) after a 3 min stimulus-free baseline period. Twenty-four hours after the final extinction session, rats received a single, weak unsignaled reminder shock (2 sec, 0.5 mA) in context A after a 3-min baseline period.

On the fifth and final day of the experiment, rats received a dual-test session for extinction retrieval (context B) and fear renewal (context C). These sessions consisted of a 3 min baseline period followed by presentation of 5 CS-alone trials (30 sec ISI). Three minutes after the final CS the recording system was paused and rats were temporarily placed in a 5-gallon buck with bedding (the recording cable remained connected), allowing us to record signal from the same neurons over

both tests. During this time, the experimenters quickly changed the contextual layout of the recording chamber (i.e., swapping from context B to context C). Rats were then placed back into the recording chamber and underwent a second retrieval session in the new context. This dual testing session enabled us to record CS-elicited activity in the same single-units in both the retrieval and renewal contexts.

Extracellular single-unit activity and freezing behavior were automatically recorded with a multichannel neurophysiological recording system (OmniPlex, Plexon, Dallas, TX). Wideband signals recorded on each channel were referenced to one of two ground wires, amplified (8000×), digitized (40 kHz sampling rate), and saved on a PC for offline sorting and analysis. After high-pass filtering (600–6000 Hz), waveforms were sorted manually using 2D principal component analysis (Offline Sorter, Plexon). Only well-isolated units with a signal-to-noise ratio greater than 3 standard deviations were used in our analysis. We then imported sorted waveforms and their timestamps to NeuroExplorer (Nex Technologies, Madison, AL) for further analysis.

**Histology**. Rats were overdosed with sodium pentobarbital (Fatal Plus; 100 mg/ml, 0.5 ml) and were transcardially perfused with ice-cold saline and 10% formalin. Brains from animals in the RE muscimol experiments were extracted and stored in 30% sucrose-formalin at 4 °C. Brains from animals in the DREADD experiments were extracted and stored in 10% formalin for up to 24 h and then transferred to 30% sucrose at 4 °C for at least 48 h. Coronal brain sections (40 μm) were made on a cryostat (−20 °C). For the animals only implanted with RE cannula, brain sections were mounted on subbed slides and stained with thionin (0.25%) to visualize cannula placements. For animals expressing viruses in the mPFC, the sections were mounted on subbed slides and coverslipped using fluoromount (Diagnostic Biosystems) to visualize viral expression.

**Fos immunohistochemistry**. Rats were overdosed with sodium pentobarbital (Fatal Plus; 100 mg/ml, 0.5 ml) and were transcardially perfused with ice-cold saline and 10% formalin. Brains were extracted and stored in 10% formalin for up to 24 h and then transferred to 30% sucrose at 4 °C for at least 48 h. Coronal brain sections (40 μm) were made on a cryostat (−20 °C). Brain sections were washed three times in TBST and then were incubated in 0.3% $H_2O_2$ for 15 min. The tissue was washed in TBS three times and was incubated in rabbit anti-c-Fos primary antibody (1:1000; Millipore) overnight. Brain tissue was washed three times in TBS followed by a 1-h incubation in a biotinylated goat anti-rabbit secondary antibody (1:1000; Jackson Immunoresearch), amplification with the avidin biotin complex at 1:1000 (ABC; Vector labs), and visualization with 3,3′-diaminobenzidine (DAB) + nickel ammonium sulfate to yield a purple/black nuclear reaction product. Stained brain sections were mounted on subbed slides, coverslipped with Permount, and stored at room temperature until photographed using a Zeiss microscope (Axio Imager).

**Image analysis**. To quantify c-fos expression, two images were taken at different A/P levels (1.9 and 2.3 from bregma) at 10X magnification (895 μm × 670 μm; 0.596 mm$^2$) of the midline thalamus. The number of Fos-positive neurons within each image and brain region were averaged and divided by the surface area to reveal the number of c-fos cells/mm$^2$.

**Data analysis**. For the RE muscimol experiments, 16 out of 152 rats were excluded from the analysis because RE cannula were misplaced or the animals did not complete the experiment. This yielded the following group sizes: encoding experiment (SAL, $n = 14$; MUS, $n = 11$), retrieval experiment (MUS = 15,Sal = 16), state-dependent experiment (SAL-SAL, $n = 20$; MUS-MUS, $n = 20$; No-ext, $n = 23$) and context-independent retrieval experiment (COND, $n = 6$; NOVEL, $n = 7$). For the intersectional DREADD experiments, 14 of 56 rats had incomplete or unilateral mPFC expression of AAV-hM4Di. This yielded the following group sizes: encoding experiment (SAL, $n = 19$, CNO, $n = 15$), retrieval experiment ($n = 7$). For the terminal DREADD experiments, 28 out of 80 rats had incomplete or unilateral mPFC expression of AAV-hM4Di and/or had misplaced RE cannulae and were excluded from the analyses. This yielded the following group sizes: Blank-GFP, $n = 17$; PL + IL DREADD, $n = 14$; PL DREADD, $n = 12$; IL DREADD, $n = 9$. All freezing data represent freezing behavior during the interstimulus intervals (ISIs). The data were analyzed using analysis of variance (ANOVA), and post-hoc comparisons in the form of Fisher's protected least significant difference (PLSD) tests were performed after a significant overall $F$ ratio in the ANOVA. For some analyses, paired or unpaired $t$ tests were used. All the data are represented as means ± s.e.m.s.

## Data availability
The data from these experiments are available from the corresponding author upon request.

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

## Acknowledgements

This work was supported by the National Institutes of Health (R01MH065961 and R01MH117852 to S.M.; F31MH112208 to T.F.G.), a McKnight Foundation Memory and Cognitive Disorders Award (S.M.), and a Brain & Behavioral Research Foundation Distinguished Investigator Grant (S.M.).

## Author contributions

S.M., K.R.R. and J.J. designed experiments, analyzed data, and wrote the manuscript. J.J. and K.R. performed the data collection. K.R.R. and M.R.P. performed the electrophysiological experiments. T.F.G. and S.M. analyzed the electrophysiological data. S.M. supervised the research.

## Additional information

**Competing interests:** The authors declare no competing interests.

