## [Peer Review File · Nature Communications]

Reviewers' Comments:

Reviewer #1:

Remarks to the Author:

In their manuscript "Prefrontal projections to the thalamic nucleus reuniens mediate fear extinction", Jin and colleagues conducted a series of elegant behavioral manipulations that demonstrate a role for the nucleus reuniens (NR) in the acquisition and retrieval of extinction memory. This adds to recent evidence from work by Wei Xu and Thomas Sudhof that demonstrated a critical role for the NR in suppressing fear responses to non-fearful context, thus controlling discrimination. To further this observation, Jin et al., now demonstrate that the NR is also involved in suppressing fearful responses to an auditory cue within an extinction (but not a non-extinction) context. By unmasking this important role for the NR in extinction memory formation and retrieval, the authors present an important contribution that supports a role for the NR in two opposite memory-related phenomena: discrimination and generalization.

The work presented here is exciting and of broad interest, given the therapeutic benefits of extinction training for human mental illnesses such as PTSD. Furthermore, the logic behind the experimental design is sound and the manuscript is easy to read and follow. However, I feel that as it stands the manuscript doesn't do enough to support its overarching message. For example, the title suggests that the authors have conclusively demonstrated a role for projections arising in the medial prefrontal cortex (mPFC) to the NR in fear extinction. However, although one may be tempted to admit their retrograde (CAV-Cre) mPFC inhibitory approach as sufficient evidence, it is not clear whether these same neurons in the mPFC innervate other brain targets, such as the amygdala or the neighboring paraventricular thalamus which has also been implicated in fear. This is my main issue with the paper, the authors clearly demonstrate that they have the ability to incorporate viral technology and stereotaxic injections in their work, yet they don't seem to be taking full advantage of this to present a more complete picture. Here are a few specific points that the authors should address:

1) In their discussion, the authors appropriately mention that the NR's role in extinction -as for the previous study with contextual fear memory- may be explained by an overall role for NR in contextual memories. However, one experiment that the authors should do to test this prediction is to condition rats to two independent CS (i.e. CS1 and CS2), and extinguish fear for only one of them in a context-independent manner. It would be interesting to see if NR silencing also recues freezing to the extinguished CS, despite no change in contextual information.

2) It appears that there is robust cFos expression in NR during renewal (Figure 4B). I am surprised by this result because silencing NR doesn't appear to have any effect on renewal (Figure 2). Based on this result one can only wonder if the neurons that are activated in the extinction context are the same as those that are activated in the non-extinction context. To answer this question the authors would benefit from in vivo electrophysiological experiments to record NR neurons over the two contexts.

3) Within the mPFC, the prelimbic (PL) and infralimbic (IL) cortices have been shown to have opposite roles in fear regulation, with PL driving fear and IL driving extinction (suppressing fear). However, the authors make no distinction for these two areas in their story. For example, muscimol in NR affects cFos expression in mPFC. Is this specific to one area of the mPFC or just mPFC in general. The authors should quantify PL and IL separately.

4) Why is there a bump in freezing during the mPFC CNO experiment? Such effect is not observed in the NR silencing experiments, which suggests that the broad silencing of the deep layers of mPFC is

qualitatively different than the NR manipulations. Similarly, it appears that the mPFC silencing also increased freezing in the renewal context. What is the explanation for this? Based on the literature, one could speculate that PL and IL projections to the NR may have opposite effects. The authors should address this experimentally. One way of doing this would be to inject a silencing opsin like halorhodopsin in either PL or IL and do optogenetic silencing of terminals over NR. This manipulation is more specific than the DREADD experiment. But, more importantly, the result may be quite exciting. The authors shouldn't ignore this opportunity to disentangle potential segregation of mPFC inputs to NR.

5) As I mentioned above, we know that deep layers mPFC neurons project to various midline thalamic regions. As such, the DREADD manipulations cannot be accepted in as a projection-specific manipulation. The same mPFC neurons projecting to NR may be projecting to other known fear regulators like the paraventricular thalamus, which could explain why the results obtained with this the mPFC DREADD experiments differ from those of the NR silencing. To investigate this further, the authors should conduct a dual retrograde tracing experiment to determine if neurons projecting to NR and the paraventricular nucleus overlap.

Minor:

Many experiments for which statistical significance is stated, have no asterisks on figures themselves. For example quantification in Figure 5B, and also the last panel in Figure 7.

Reviewer #2:

Remarks to the Author:

In the manuscript entitled 'Prefrontal projections to the thalamic nucleus reuniens mediate fear extinction', Jin et al. found inactivation of thalamic nucleus reuniens with local infusion of muscimol could impair formation and expression of fear extinction. They revealed that impairment was accompanied by reduced c-Fos expression in both medial frontal cortex and ventral hippocampus. They further verified specifically inactivated prefrontal projections to the nucleus reuniens led to similar impairment.

This study brings new insights to understand the mechanism of fear extinction. Overall, the experiments were well designed and the manuscript was well written. The major conclusion was well supported by their experiment results. Nevertheless, there are few major and minor concerns, which are listed below;

Major:

1. On Page 7 and Figure 3, the experiment determines the muscimol-induced extinction impairments are not state-dependent is questionable. In order to exclude the possible drug-shift induced fear renewal in the extinction retrieval testing, they did two drug infusions to make same context for the extinction and retrieval. However, the first inactivation was supposed to impair extinction already, which made the second drug infusion useless. It is not valid to claim the impairment of extinction in the retrieval is caused by the second muscimol infusion.
2. In the experiment silencing of medial prefrontal cortex to nucleus reuniens projections, validation to the specificity of the DREADD virus and the effectiveness of retrograde Cre virus is better to be included.
3. As shown in the result, inactivation of RE decreased c-Fos expression in both medial prefrontal cortex and ventral hippocampus, a brief discussion about the potential mechanism is welcome.

Minor:

1. On page 4, expression in line 14-16 is not very clear.
2. On page 20, in second last line, expression 'followed by 3 min baseline' is confusing. Is the baseline before or after the tone presentation?
3. On page 20, in the last line, there is possible typo error '35 min 30s'.
4. On page 20, 21, within the behavior procedures, there are two different durations (3 minutes and 10 minutes) chosen for baselines. Is there any justification for using different baselines?
5. In Figure 1, 4, 5, 6, it's better to include scale bars for the histological pictures.
6. In Figure 4, 6B, the names for nucleus reuniens are not consistent with Figure 1 and also the text.

Reviewer #3:

Remarks to the Author:

The study by Jin et al investigates the role of nucleus reuniens (RE) in Pavlovian fear extinction. Using pharmacological and chemogenetic approaches, the authors showed that inactivation of RE cells before extinction and before post-extinction retrieval increased rats freezing responses, suggesting that the encoding and consolidation of extinction were impaired. In addition, they revealed that context-dependent fear renewal was not affected. Although the demonstration that the mPFC-RE pathway plays a role in fear extinction is somehow novel, based on the known literature these findings were to some extent expected: (i) the mPFC is required for extinction retrieval and context-dependent fear renewal (Quirk et al., 2000; Orsini et al., 2011), (ii) hippocampal (HIP) cells projecting to the mPFC are activated during context-dependent fear renewal (Jin and Maren, 2015; Wang et al., 2016) and (iii) the mPFC projects to the vHIP through the RE (Xu and Sudhof, 2013; Bokor et al, 2002; Wouterlood et al, 1990). Moreover, there are several criticisms the authors need to address and additional experiments to perform before this paper can be considered for publication. For these reasons, we believe that the present findings are more appropriate for publication in more specialized journals such as "Scientific Report", "Learning and Memory" or "Journal of Neuroscience".

MAJOR CONCERNS:

- The authors show that during renewal (ex. Fig. 2, 4, 6) rats at baseline (BL) show no freezing responses. However, one would expect freezing to be significantly increased because of contextual conditioning. A potential reason is the animal exposure for the renewal context for 35 min before freezing quantification, as stated in the methods section. The authors should at least provide some rationale for the 35 min exposure and show data about this dynamics in a supplementary figure.
- In all the behavioral experiments using Muscimol or CNO the authors should provide more detailed locomotion tests to confirm no side effects of their manipulations. Indeed, as they do not show any data about the dynamics on rats' behaviors (speed, distance travelled, thigmotaxis, ...) during baseline, they cannot conclude that there is no locomotor side effect.
- In Figure 4 the authors assess whether RE cells were activated during both encoding and retrieval of extinction. Moreover, in Figure 5 they investigate if the inactivation of RE cells leads to reduce activity of mPFC and vHIP. In order to test these hypotheses, they quantified c-fos protein expression levels by providing a number of immunoreactive cells, which is not enough to support the authors main conclusions. In addition to these rough numbers, the authors should provide density or at least normalization of the surface analyzed. Also, the authors surprisingly did not explain how they analyzed and quantified c-fos levels. This must more be carefully described in the methods section (confocal images, stack, slice size, detection method, threshold for quantification, selection of nucleus boundaries...).

- Again, in Figure 5, the authors used c-fos analyses to demonstrate that RE Muscimol inactivation promotes c-fos expression in the mPFC and vHIP, which is a fairly indirect demonstration. It is not clear at all if RE inactivation impacts directly the number of c-fos+ cells in the vHIP and mPFC. Additional experiments, including specific tracing experiments (ctb + c-fos) are therefore required to provide such claims.
- The authors reveal opposing effects on renewal when DREADDs (Figure 6) are used instead of Muscimol (Figure 2). This is particularly confusing as it suggests some weaknesses in the methodology used throughout the manuscript. At least some detailed explanation should be provided.
- In figure 7 the authors use a within subject design whereas in previous experiments a between subject design was used. What is the rationale behind this? We suggest using the same designs across all the experiments.
- We agree that DREADDs are a very useful tool. However, the recent literature about their use is very controversial. Therefore, the authors should provide a demonstration that this tool, in their hands, really works. For example, they could provide electrophysiological recordings that cells are indeed inhibited upon CNO administration or on the contrary, replicate their results with optogenetic approaches.
- In Figure 6A, the authors used non-specific DREADDs injection, which results in the labeling of both PL and IL cells. This is very problematic given the known literature from the same group: PL is activated during context dependent fear renewal whereas IL is activated during extinction retrieval (Knapska and Maren, 2009; Wang et al., 2016). More specific experiments must be provided here.

MINOR POINTS:

- The authors indicate that rats were transported for different contexts in different colored plastic boxes. Could the authors provide a rationale for this choice? It is very likely that rats could predict based on the transport boxes to which context they will be exposed.
- In Figure 3 authors reveal that the effect of Muscimol does not depend on state-dependent processes. As this is not a crucial finding for the story but just a control, we believe this experiment should be provided as a supplementary figure.
- In Figure 4 and 5, the authors should use a low magnification picture to show the specificity of c-fos expression. Moreover, some comparisons to other structures involved and not involved in fear extinction and renewal should be provided.
- Scale bars from representative pictures are not shown.
- In most of the figures and captions the authors do not show statistical significance when it is the case, please provide this essential information to help the reader.
- Figure 8 should be provided as a supplementary Figure as it only controls for CNO effects on WT rats.

Reply to Reviewers

We thank the editor and all three reviewers for their time and insightful comments on our manuscript. We have completely revised the manuscript based on this feedback and have included three new experiments requested by the reviewers. These experiments establish that 1) the nucleus reuniens (RE) has a ubiquitous role in extinction retrieval independent of the extinction procedure used to reduce CS freezing, 2) single-units in the RE respond more vigorously to extinguished CSs when fear is inhibited (in the extinction context), and 3) projection-specific manipulations of IL and PL terminals in RE reveal similar functions of these inputs in extinction retrieval--inactivating either projection impaired extinction retrieval. These new experiments have greatly improved the manuscript and strengthened our conclusion that medial prefrontal cortex projections to the RE mediate extinction learning and memory. Below, we address each reviewer's comments individually (reviewer comments are numbered and listed in **bold**). We have underlined the text where we added new experiments and have indicated changes to the main text in *italic bold*.

Reviewer #1:

The reviewer notes that we have used “elegant behavioral manipulations” to demonstrate a role for the RE in the acquisition and retrieval of extinction memory. The reviewer lauds our work as “exciting and of broad interest” and making an “important contribution” to the field. The reviewer's main concern, however, was that we did not specifically implicate mPFC projections to RE in these processes. We have taken this criticism to heart and have now added a new terminal inactivation experiment to address this issue. A reply to each of the reviewer's concerns is noted below:

1) In their discussion, the authors appropriately mention that the NR's role in extinction -as for the previous study with contextual fear memory- may be explained by an overall role for NR in contextual memories. However, one experiment that the authors should do to test this prediction is to condition rats to two independent CS (i.e. CS1 and CS1), and extinguish fear for only one of them in a context-independent manner. It would be interesting to see if NR silencing also recues freezing to the extinguished CS, despite no change in contextual information.

This is an interesting point and we have conducted a new experiment to determine whether the RE mediates extinction retrieval when extinction occurs in the same context as conditioning. Rats underwent a standard fear conditioning procedure, and then we extinguished one group of animals in the in a novel context (as in the paper) and extinguished another group of animals in the conditioning context. Although both extinction procedures produce context-dependent extinction, extinguishing rats in a novel context typically produces more context-dependency than extinguishing rats in the conditioning context (i.e., ABA/ABC renewal is typically greater than AAB renewal). After extinction, we then examined whether RE inactivation would differentially affect extinction retrieval under these conditions. We found that RE inactivation impaired the retrieval of fear memory independent of the context in which the CS was extinguished (*see page 8, lines 177-202; Supplemental Figure 2*). This reveals that the RE has a ubiquitous role in the inhibition of fear to an extinguished CS, and that extinction does not need to occur in a novel context to recruit the RE.

2) It appears that there is robust cfos expression in NR during renewal (Figure 4B). I am surprised by this result because silencing NR doesn't appear to have any effect on renewal (Figure 2). Based on this result one can only wonder if the neurons that are activated in the extinction context are the same as those that are activated in the non-extinction context. To answer this question the authors would benefit from in vivo electrophysiological experiments to record NR neurons over the two contexts.

It is possible that the increased Fos expression we observed after both retrieval and renewal tests is due to the low temporal resolution of Fos. At the reviewer's suggestion, we have performed a new electrophysiological experiment in which we recorded from RE neurons during both retrieval and renewal sessions. In this

experiment, we used a within-subject design that allowed us to record the response of individual RE neurons to the same CS presented in the extinction context (where fear is low) and in the renewal context (where fear is high). We found that RE neurons show greater CS-elicited firing in the extinction context compared to a renewal context. This indicates that RE neurons selectively increase their firing to CSs that suppress fear and confirm the role for these neurons in fear inhibition (*see page 10, lines 222-245; Figure 2G-J*)

3) Within the mPFC, the prelimbic (PL) and infralimbic (IL) cortices have been shown to have opposite roles in fear regulation, with PL driving fear and IL driving extinction (suppressing fear). However, the authors make no distinction for these two areas in their story. For example, muscimol in NR affects cfos expression in mPFC. Is this specific to one area of the mPFC or just mPFC in general. The authors should quantify PL and IL separately.

We have now added a new experiment to address the independent contributions of PL or IL projections to the RE in extinction memory retrieval. We injected AAVs expressing inhibitory DREADDs into the PL, IL, or both and then inactivated mPFC terminals in the RE with local infusions of CNO prior to an extinction retrieval test (*see Figure 4 and Supplemental Figure 3*). This experiment revealed that inactivating terminals from IL, PL, or both produced similar impairments in extinction memory retrieval. Although IL and PL have different roles in fear expression under some conditions (as the reviewer notes), the terminal inactivation experiment suggests that both IL and PL projections to RE are involved in fear inhibition after extinction (*see page 12, lines 287-316; Figure 4 and Supplemental Figure 3*). Indeed, there is electrophysiological evidence from numerous studies that both IL and PL neurons increase their firing to extinguished CSs (reviewed in Giustino and Maren, 2015). This suggests that the opposing role IL and PL neurons play in extinction are likely mediated by efferent projections to other subcortical targets, such as the amygdala.

4) Why is there a bump in freezing during the mPFC CNO experiment? Such effect is not observed in the NR silencing experiments, which suggests that the broad silencing of the deep layers of mPFC is qualitatively different than the NR manipulations. Similarly, it appears that the mPFC silencing also increased freezing in the renewal context. What is the explanation for this? Based on the literature, one could speculate that PL and IL projections to the NR may have opposite effects. The authors should address this experimentally. One way of doing this would be to inject a silencing opsin like halorhodopsin in either PL or IL and do optogenetic silencing of terminals over NR. This manipulation is more specific than the DREADD experiment. But, more importantly, the result may be quite exciting. The authors shouldn't ignore this opportunity to disentangle potential segregation of mPFC inputs to NR.

The reviewer correctly notes that there was an increase in freezing in the extinction session for animals treated with CNO during extinction that was not evident in the RE inactivation experiment. This “bump” in freezing occurred because the CNO experiment was run with three extinction sessions, because there was poor within-session extinction in the control animals after the first two extinction sessions. The data shown in the figure were from the third session of extinction, and therefore reflected the impairment in extinction produced by CNO administration on the first two days of extinction. We have now revised the figure (*see Figure 3C*) to show the data from first extinction session in which the animals were first administered CNO (a session that matches the data shown for the RE inactivation experiment). During the first extinction session, CNO administration did not increase freezing in the first extinction block.

With respect to possibility that PL and IL input to RE have different function in the extinction, we have added a new experiment that addressed this possibility (see the response to #3). Inhibition of IL or PL terminals (or both) increased freezing to an extinguished CS and similarly impaired the expression of extinction.

5) As I mentioned above, we know that deep layers mPFC neurons project to various midline thalamic regions. As such, the DREADD manipulations cannot be accepted in as a projection-specific manipulation. The same mPFC neurons projecting to NR may be projecting to other known fear

regulators like the paraventricular thalamus, which could explain why the results obtained with this the mPFC DREADD experiments differ from those of the NR silencing. To investigate this further, the authors should conduct a dual retrograde tracing experiment to determine if neurons projecting to NR and the paraventricular nucleus overlap.

The reviewer is correct in noting that DREADD manipulations of mPFC projection neurons are not necessarily specific to RE (despite our use of an intersectional approach), because those mPFC neurons might have collaterals to other subcortical targets. It is possible that collateral projections from the mPFC to different thalamic (or other subcortical targets) targets mediate different processes associated with the encoding and retrieval of extinction memories. As described above, we have now added an additional experiment in which we specifically manipulated mPFC terminals in the RE using local infusions of CNO in the RE in animals in which we injected AAVs expressing inhibitory DREADDs in the IL, PL, or both (*see page 12, lines 287-316; Figure 4 and Supplemental Figure 3*). These experiments reveal that inhibition of either IL or PL terminals in RE increase freezing to an extinguished CS and impair extinction retrieval.

Minor:

1) Many experiments for which statistical significance is stated, have no asterisks on figures themselves. For example, quantification in Figure 5B, and also the last panel in Figure 7.

We have updated the figures and indicated the major significant differences with asterisk marks which are central to our interpretations.

Reviewer #2:

This reviewer noted that our study “brings new insights to understand the mechanism of fear extinction”. The reviewer also noted that our experiments were well designed and that the major conclusions are supported by the experimental results. They had similar concerns to Reviewer #1 regarding the projection specificity of our intersectional (CAV2-Cre) manipulation, which we have now addressed with new experiments. We respond to the reviewer’s concerns below:

1. On Page 7 and Figure 3, the experiment determines the muscimol-induced extinction impairments are not state-dependent is questionable. In order to exclude the possible drug-shift induced fear renewal in the extinction retrieval testing, they did two drug infusions to make same context for the extinction and retrieval. However, the first inactivation was supposed to impair extinction already, which made the second drug infusion useless. It is not valid to claim the impairment of extinction in the retrieval is caused by the second muscimol infusion.

In the RE inactivation experiments, we examined conditional freezing during a retrieval test in which the “drug state” was different from that during encoding. That is, in one experiment we inactivated the RE with muscimol (MUS) during extinction and tested the animals under saline; in another experiment, we extinguished the animals under saline and tested them after RE inactivation with MUS. Because extinction is context-dependent, these shifts in drug state from the extinction session to the retrieval test might have themselves caused a return of conditional freezing to the CS (i.e., a renewal effect; Bouton et al 1990, ref 29 in main text).

To test this possibility, we conducted a control experiment (Figure 1F) in which we compared animals that were both extinguished and tested after RE inactivation with animals that were either not extinguished or extinguished and tested after saline infusions. If the deficits in extinction produced by RE inactivation are state-dependent, than animals in the MUS-MUS condition (state matched between extinction and testing) should express extinction and be no different from animals extinguished and tested after saline infusions. However, we found that the MUS-MUS animals showed increased freezing on test relative to SAL-SAL controls, and did not

differ from non-extinguished animals (*see page 7, lines 153-176; Figure 1F*). This reveals that the deficits in extinction associated with RE inactivation during extinction or retrieval are not due to state-dependent generalization deficits. As the reviewer points out, the high levels of freezing observed in the MUS-MUS animals are most parsimoniously explained by a failure to encode and/or retrieve the extinction memory.

2. In the experiment silencing of medial prefrontal cortex to nucleus reunites projections, validation to the specificity of the DREADD virus and the effectiveness of retrograde Cre virus is better to be included.

We have validated the function of the viruses used in our experiments in a recently published paper (Marek et al 2018; ref 27 in main text). As shown in the figure to the right (Supplementary Figure 5 from Marek et al., 2018),

mPFC neurons exhibited dose-dependent decreases in spontaneous firing (measured using extracellular array recordings) after the systemic administration of CNO in intact animals infected with AAV-hM4Di in mPFC; CNO had no effect in animals infected with a blank control virus.

3. As shown in the result, inactivation of RE decreased c-Fos expression in both medial prefrontal cortex and ventral hippocampus, a brief discussion about the potential mechanism is welcome.

We have removed the experiment showing the RE inactivation reduces retrieval-induced Fos expression in the mPFC and hippocampus, because we thought that this was beyond the scope of the paper which we have focused on mPFC efferents to RE in extinction encoding and retrieval. We are now pursuing additional experiments that will examine RE efferents to mPFC and hippocampus in these processes, and feel that those data will stand alone as a separate report. That said, we have followed up upon the RE Fos data with additional electrophysiological experiments to better understand the function of RE neurons in extinction retrieval. Those experiments are described above (Reviewer #1, point #3).

Minor:

1. On page 4, expression in line 14-16 is not very clear.

We have made edits to that sentence to make it clearer.

2. On page 20, in second last line, expression ‘followed by 3 min baseline’ is confusing. Is the baseline before or after the tone presentation?

The 3-minute baseline is prior to the onset of the first tone CS. We have made edits to the manuscript to make it clearer (*see page 20; line 473*)

3. On page 20, in the last line, there is possible typo error ‘35 min 30s’.

Our extinction session involves a 3 min BL followed by 45 tone trials (10 sec each) with a 30-sec interstimulus interval a 3-min period after the last CS. This adds up to a total of 35 mins and 30 seconds. For exposure, we place the animals in renewal context for the same duration as extinction session.

4. On page 20, 21, within the behavior procedures, there are two different durations (3 minutes and 10 minutes) chosen for baselines. Is there any justification for using different baselines?

We included the longer, 10-min baseline during the retrieval tests because RE has been previously been shown to have a role in the generalization of fear across contexts (Xu et al 2013; ref 24 in main text). Because we

tested animals in different contexts from those in which they were conditioned, we wanted to detect any increases in fear generalization produced by RE inactivation. We thought a longer baseline period would provide a more sensitive test of generalization; that said, RE inactivation did not affect baseline freezing in any of the experiments.

5. In Figure 1 , 4, 5, 6, it's better to include scale bars for the histological pictures.

We have added the scale bar for the histological sections.

6. In Figure 4, 6B, the names for nucleus reuniens are not consist with Figure 1 and also the text.

We have fixed the error to make it more consistent across the figures and text.

Reviewer #3:

The reviewer described our study as novel, but suggested that other studies of HPC-mPFC circuits in the context-dependent expression of extinction anticipate a role for the RE in extinction. However, previous work on these circuits (e.g., Marek et al., 2018) has shown that HPC-mPFC circuits promote fear renewal by inhibiting extinction retrieval. In the present work, we show the converse: *projections from the mPFC to RE promote extinction retrieval and inhibit the expression of conditioned freezing.* This is an important and novel observation (which is echoed by reviewers #1 and #2), because current circuit models of extinction posit that mPFC projections to the amygdala engage inhibitory networks to suppress fear. Indeed, we show for the first time that the inhibition of conditioned fear after extinction is mediated by mPFC projections to RE that inhibit encoding and retrieval of the extinction memory. The reviewer had several other concerns that we address below:

1) The authors show that during renewal (ex. Fig. 2, 4, 6) rats at baseline (BL) show no freezing responses. However, one would expect freezing to be significantly increased because of contextual conditioning. A potential reason is the animal exposure for the renewal context for 35 min before freezing quantification, as stated in the methods section. The authors should at least provide some rational for the 35 min exposure and show data about this dynamics in a supplementary figure.

As the reviewer notes, baseline freezing in the conditioning context during the renewal tests is low. The reviewer is correct that freezing is low because we extinguish contextual freezing to the conditioning context with a 35-min exposure session on the extinction day. This is a standard procedure in the field that is explicitly performed to reduce contextual freezing that would otherwise interfere with CS freezing during the renewal test. The context exposure data for animals assigned to the MUS or SAL conditions in the two experiments are shown here (the animals were not on drug during these sessions). Animals showed substantial extinction to the conditioning context during these sessions, and this accounted for the low baseline freezing during renewal testing (which was the reason we exposed them to the conditioning context).

2) In all the behavioral experiments using muscimol or CNO the authors should provide more detailed locomotion tests to confirm no side effects of their manipulations. Indeed, as they do not show any data about the dynamics on rats' behaviors (speed, distance travelled, thigmotaxis, ...) during baseline, they cannot conclude that there is no locomotor side effect.

We have used overall motor activity to determine whether there are non-specific effects of MUS or CNO on behavior that might have accounted for differences in conditional freezing. Motor activity in our apparatus is measured by the physical displacement of the conditioning chamber which is detected by a force transducer (load-cell) under the chamber. This method aggregates activity produced by several behaviors including grooming, rearing, and locomotion. Our activity monitoring system does not collect positional information, so we do not have the capacity to measure speed, distance traveled, or location.

To determine whether muscimol infusions in RE or systemic CNO administrations produced their effects by non-specifically decreasing motor activity, we assessed baseline freezing and motor activity during a 10-minute baseline prior to the extinction retrieval tests. The levels of freezing and motor activity were comparable during this test and this is reported in the manuscript (*see page 7, line 150, page 11, line 268; page 12, line 284*). The figure on the right shows motor activity during the 10-min baseline periods prior to RE inactivation with muscimol or systemic CNO administration. The data represent load-cell voltages sampled at 5Hz of chamber displacement during the test for all the animals included in the analysis each group; 1-min averages of these values is shown below each plot. The activity data show that there are no differences in drug-treated animals in their overall motor activity across the 10-min test, revealing that nonspecific decreases in activity do not account for the increased freezing observed to presentation of the CS after the baseline (the overall decrease in activity across the test occur as animal's habituate their exploratory activity of the context). In recently published work, we also failed to observe non-specific motor effects of intracranial muscimol or systemic CNO (Marek et al., 2018). We believe these data support the conclusion that pharmacological inactivation of RE or its prefrontal afferents increases freezing by impairing the retrieval of an extinction memory.

3) In Figure 4 the authors assess whether RE cells were activated during both encoding and retrieval of extinction. Moreover, in Figure 5 they investigate if the inactivation of RE cells leads to reduce activity of mPFC and vHIP. In order to test these hypotheses, they quantified c-fos protein expression levels by providing a number of immunoreactive cells, which is not enough to support the authors main conclusions. In addition to these rough numbers, the authors should provide density or at least normalization of the surface analyzed. Also, the authors surprisingly did not explain how they analyzed and quantified c-fos levels. This must more be carefully described in the methods section (confocal images, stack, slice size, detection method, threshold for quantification, selection of nucleus boundaries...).

We have made extensive revisions in the methods sections to make the quantification of the c-fos experiment clearer. We have added a section called “Image analysis” wherein we mentioned the surface area of the brain region we counted. In addition to that, we normalized the c-fos counts by dividing the raw cell counts to surface area and represented it as cells/mm² which is now comparable across brain regions (*see page 24, Lines 574-578, Figure 2*).

4) Again, in Figure 5, the authors used c-fos analyses to demonstrate that RE Muscimol inactivation promotes c-fos expression in the mPFC and vHIP, which is a fairly indirect demonstration. It is not clear at all if RE inactivation impacts directly the number of c-fos+ cells in the vHIP and mPFC. Additional experiments, including specific tracing experiments (ctb + c-fos) are therefore required to provide such claims.

As described above (Reviewer #2, point #3), we have now removed the experiment showing the RE inactivation reduces retrieval-induced Fos expression in the mPFC and hippocampus, because we thought that this was beyond the scope of the paper which we have focused on mPFC efferents to RE in extinction encoding and retrieval. We are now pursuing additional experiments that will examine RE efferents to mPFC and hippocampus in these processes, and feel that those data will stand alone as a separate report. That said, we have followed up upon the RE Fos data with additional electrophysiological experiments to better understand the function of RE neurons in extinction retrieval. Those experiments are described above (Reviewer #1, point #3).

5) The authors reveal opposing effects on renewal when DREAADs (Figure 6) are used instead of Muscimol (Figure 2). This is particularly confusing as it suggests some weaknesses in the methodology used throughout the manuscript. At least some detailed explanation should be provided.

In the present study, we used muscimol to directly inactivate the RE during extinction encoding and retrieval, and CNO to inactivate mPFC neurons projecting to RE during these processes. We have also added a new experiment with local CNO infusions into RE to manipulate mPFC terminals selectively. As the reviewer notes, although we obtained identical outcomes with the two manipulations on extinction encoding and retrieval, the results of these manipulations differed slightly on the renewal test. Specifically, CNO-mediated inhibition of mPFC neurons projecting to RE during extinction led an increase in freezing during the renewal test, whereas MUS inactivation of the RE did not. This suggests that CNO inactivation of PFC->RE projection neurons produces a greater extinction encoding impairment than MUS inactivation of RE. However, it is also possible that the different outcomes on the renewal test were due to the relatively lower level of renewal in the control animals in the CNO experiment, which would have made increased renewal in the drug-treated animals easier to detect. But the somewhat different outcomes on the renewal test do not detract from the similar effect these different manipulations had on extinction retrieval: In both cases inactivating RE or its projections from mPFC robustly increased freezing during the extinction retrieval test, suggesting that these manipulations impair encoding and retrieval of the extinction memory.

6) In Figure 7 the authors use a within-subject design whereas in previous experiments a between subject design was used. What is the rationale behind this? We suggest using the same designs across all the experiments.

We used within-subject designs to assess CNO effects on extinction retrieval (see Figure 3C and Figure 4C) in a manner that allowed each animal served as its own control (i.e., each animal was tested in a counterbalanced manner with either CNO or saline). This design is particularly powerful in the viral experiments given considerable individual differences in the extent of infection from animal to animal. We also used within-subject designs to examine drug effects on freezing during the retrieval and renewal tests. That, animals were tested to the extinguished CS in both the extinction or conditioning contexts after either drug or saline administration.

As the reviewer notes, for the extinction encoding experiments we used a between-subject design in which separate groups of rats were administered either drug or saline. This approach was necessary, because we were interested in how RE inactivation affected extinction learning using a single CS—we could not both extinguish under drug and saline in the same animal and understand how those manipulations differently affect extinction learning. An alternative within-subject design would require that the animals undergo conditioning with two different CSs. One of those would undergo extinction under drug and the other under saline prior to drug-free retrieval tests to both CSs. The difficulty with this sort of design is that it changes the nature of the conditioning procedure (twice as many conditioning trials), yields generalized fear and extinction across the CSs, and increases the overall number of extinction trials. For this reason, the encoding experiments were conducted with the between-subject design that matches the conditioning and extinction procedures of the other experiments.

7) We agree that DREADDs are a very useful tool. However, the recent literature about their use is very controversial. Therefore, the authors should provide a demonstration that this tool, in their hands, really works. For example, they could provide electrophysiological recordings that cells are indeed inhibited upon CNO administration or on the contrary, replicate their results with optogenetic approaches.

As described above (Reviewer #2, point # 2), we have validated the function of the viruses used in our experiments in a recently published paper (Marek et al 2018; ref 27 in main text). As shown in the figure to the right (Supplementary Figure 5 from Marek et al., 2018), mPFC neurons exhibited dose-dependent decreases in spontaneous firing (measured using extracellular array recordings) after the systemic administration of CNO in intact animals infected with AAV-hM4Di in mPFC; CNO had no effect in animals infected with a blank control virus.

mPFC neurons exhibited dose-dependent decreases in spontaneous firing (measured using extracellular array recordings) after the systemic administration of CNO in intact animals infected with AAV-hM4Di in mPFC; CNO had no effect in animals infected with a blank control virus.

8) In Figure 6A, the authors used non-specific DREADDs injection, which results in the labeling of both PL and IL cells. This is very problematic given the known literature from the same group: PL is activated during context dependent fear renewal whereas IL is activated during extinction retrieval (Knapska and Maren, 2009; Wang et al., 2016). More specific experiments must be provided here.

As described above (Reviewer #1, point #2), we have added a new experiment to address the independent contributions of PL or IL projections to the RE in extinction memory retrieval. We injected AAVs expressing inhibitory DREADDs into the PL, IL, or both and then inactivated mPFC terminals in the RE with local infusions of CNO prior to an extinction retrieval test. This experiment revealed that inactivating terminals from IL, PL, or both produced similar impairments in extinction memory retrieval. Although IL and PL have different roles in fear expression under some conditions (as the reviewer notes), the terminal inactivation experiment suggests that both IL and PL projections to RE are involved in fear inhibition after extinction (*see page 12, lines 287-316; Figure 4 and Supplemental Figure 3*). Indeed, there is electrophysiological evidence from numerous studies that both IL and PL neurons increase their firing to extinguished CSs (reviewed in Giustino and Maren, 2015). This suggests that the opposing role IL and PL neurons play in extinction are likely mediated by efferent projections to other subcortical targets, such as the amygdala.

MINOR POINTS:

- **The authors indicate that rats were transported for different contexts in different colored plastic boxes. Could the authors provide a rationale for this choice? It is very likely that rats could predict based on the transport boxes to which context they will be exposed.**

Yes, the animals can use the transport cues to predict the different contexts that would enter. This is a standard practice in many labs that promotes discrimination between the various contexts used in the experiments.

- **In Figure 3 authors reveal that the effect of Muscimol does not depend on state-dependent processes. As this is not a crucial finding for the story but just a control, we believe this experiment should be provided as a supplementary figure.**

We have kept this experiment in the main figure (Figure 1) describing the muscimol experiments because we feel that it is central to the conclusion that inactivation of the RE impairs extinction memory encoding and retrieval (rather than producing a state-dependent decrement in generalization of extinction).

- In Figure 4 and 5, the authors should use a low magnification picture to show the specificity of c-fos expression. Moreover, some comparisons to other structures involved and not involved in fear extinction and renewal should be provided.

Below are lower magnification images showing Fos staining in the midline thalamus for all the groups in the present study. We restricted our analysis to RE because previous studies (e.g., Do Monte et al., 2015) have characterized other midline nuclei (including the paraventricular nucleus and mediodorsal nuclei) in fear conditioning.

- Scale bars from representative pictures are not shown.

We have updated the figures to show scale bars.

- In most of the figures and captions the authors do not show statistical significance when it is the case, please provide this essential information to help the reader.

We have made changes to figures to indicate the significant differences in the figures.

- Figure 8 should be provided as a supplementary Figure as it only controls for CNO effects on WT rats.

We have removed the control CNO experiment as we have included blank viruses in the terminal inactivation experiments—these experiments indicate that CNO alone does not affect extinction retrieval.

Reviewers' Comments:

Reviewer #1:

Remarks to the Author:

Through a series of additional experiments and on-point responses, the authors have thoroughly and elegantly addressed this reviewer's concerns. I have no further comments.

Reviewer #2:

Remarks to the Author:

This revised manuscript includes new experiment results to address one of reviewer's major concerns, two parts of mPFC (PL and IL) may have different roles in fear expression. The results are informative and should eliminate doubts. In the revised manuscript, authors also excluded inconclusive c-Fos results on vHIP. In response to reviewer's request, description of c-Fos experiment method was added. Taking together, these modifications significantly improve the manuscript. Main conclusion on RE's critical roles in both fear extinction learning and retrieval is well supported by current data set . Since there are a couple of obvious grammar errors and some unclear expressions in the manuscript, I would suggest the authors to do a careful proof reading before publication.

Reviewer #3:

Remarks to the Author:

The authors have performed a number of new experiments and have addressed all the points I have raised. The manuscript is significantly improved. I have no further comments